# Structures of PKA–phospholamban complexes reveal a mechanism of familial dilated cardiomyopathy

**Juan Qin**[1], **Jingfeng Zhang**[2], **Lianyun Lin**[1], **Omid Haji-Ghassemi**[3], **Zhi Lin**[4], **Kenneth J Woycechowsky**[1], **Filip Van Petegem**[3], **Yan Zhang**[1]*, **Zhiguang Yuchi**[1,5]*

[1]Tianjin Key Laboratory for Modern Drug Delivery & High-Efficiency; Collaborative Innovation Center of Chemical Science and Engineering; School of Pharmaceutical Science and Technology, Tianjin University, Tianjin, China; [2]Wuhan Institute of Physics and Mathematics, Chinese Academy of Sciences, Wuhan, China; [3]Department of Biochemistry and Molecular Biology, The Life Sciences Centre, University of British Columbia, Vancouver, Canada; [4]School of Life Sciences, Tianjin University, Tianjin, China; [5]Department of Molecular Pharmacology, Tianjin Medical University Cancer Institute & Hospital; National Clinical Research Center for Cancer; Key Laboratory of Cancer Prevention and Therapy, Tianjin; Tianjin's Clinical Research Center for Cancer, Tianjin, China

*For correspondence:
yan.zhang@tju.edu.cn (YZ);
yuchi@tju.edu.cn (ZY)

**Competing interest:** The authors declare that no competing interests exist.

**Abstract** Several mutations identified in phospholamban (PLN) have been linked to familial dilated cardiomyopathy (DCM) and heart failure, yet the underlying molecular mechanism remains controversial. PLN interacts with sarco/endoplasmic reticulum $Ca^{2+}$-ATPase (SERCA) and regulates calcium uptake, which is modulated by the protein kinase A (PKA)-dependent phosphorylation of PLN during the fight-or-flight response. Here, we present the crystal structures of the catalytic domain of mouse PKA in complex with wild-type and DCM-mutant PLNs. Our structures, combined with the results from other biophysical and biochemical assays, reveal a common disease mechanism: the mutations in PLN reduce its phosphorylation level by changing its conformation and weakening its interactions with PKA. In addition, we demonstrate that another more ubiquitous SERCA-regulatory peptide, called another-regulin (ALN), shares a similar mechanism mediated by PKA in regulating SERCA activity.

## Editor's evaluation

As part of the "flight or fight" response, protein kinase A phosphorylates phospholamban (PLN), thereby relieving a tonic inhibition of the endo/sarco-plasmic reticulum calcium pump, which results in an increased force of cardiac contraction. This paper describes results on the association between protein kinase A and peptides corresponding to wild-type PLN (residues 8-22) and peptides bearing mutations (R9C and A11E) which in the context of full-length PLN (52 residues) result in dilated cardiomyopathy. The work will be of interest to investigators of mechanisms of substrate recruitment by protein kinases, and particularly to those who are trying to understand the mechanisms of familial dilated cardiomyopathy.

## Introduction

Dilated cardiomyopathy (DCM) is the most common type of cardiomyopathy, characterized by an enlarged heart with a decreased ejection fraction. It is a major cause of heart failure (*Dellefave and*

*McNally, 2010*), affecting 40 million people globally (*Dellefave and McNally, 2010*; *Jefferies and Towbin, 2010*; *Disease et al., 2016*). 25–35% of DCM cases have familial origin (*Kureel et al., 2013*), caused by inherited mutations in genes encoding proteins involved in muscle contraction and calcium handling, including phospholamban (PLN) (*Alves et al., 2010*; *MacLennan, 2000*; *MacLennan and Kranias, 2003*; *Kranias and Bers, 2007*).

To initiate cardiac muscle contraction, an action potential depolarizes the sarcolemma and activates the voltage-gated calcium channel, $Ca_V1.2$, which mediates $Ca^{2+}$ influx (*Bers, 2002*). The small increase in the cytosolic $Ca^{2+}$ concentration causes larger-scale calcium-induced calcium release from the intracellular sarcoplasmic reticulum (SR) stores through cardiac ryanodine receptors (RyR2) (*Fabiato, 1983*; *Santulli and Marks, 2015*). The resulting increase of the cytosolic $[Ca^{2+}]$ from 100 nM to 10 μM induces muscle contraction (*Santulli and Marks, 2015*; *Copello et al., 1997*; *Laver et al., 1995*). To relax the cardiac muscle, sarcoplasmic reticulum $Ca^{2+}$-ATPases (SERCAs) on the SR membrane couple ATP hydrolysis to the pumping of $Ca^{2+}$ back into the SR (*Bers, 2008*; *Hill and Inesi, 1982*). The rate and duration of SERCA-mediated SR calcium restoration affect the SR calcium load, which regulates the rate of muscle relaxation and the intensity of the next contraction.

PLN, an important regulator of SERCA, is a 6.2 kDa single-pass integral membrane protein that can reversibly inhibit SERCA activity by physically interacting with the calcium pump and thus regulate the contraction of cardiac muscle (*Tada et al., 1976*; *Simmerman and Jones, 1998*; *Traaseth et al., 2008*). The transmembrane domain of PLN interacts with SERCA via a conserved sequence motif (*MacLENNAN et al., 1998*). Inhibition of SERCA by PLN responds to changes in the free calcium concentration and redox environment of the cytosol. The potency of SERCA inhibition also depends on certain structural properties of PLN, such as its oligomerization state and phosphorylation levels. PLN can be phosphorylated by cAMP-dependent protein kinase A (PKA) and calmodulin-dependent protein kinase II (CaMKII) (*Tada et al., 1976*; *Simmerman et al., 1986*; *Ha et al., 2011*). During the fight-or-flight response, the activation of β-adrenergic receptor leads to the activation of PKA, which in turn phosphorylates a number of downstream targets regulating cardiac muscle contraction, including $Ca_V1.2$, RyR2, troponin, and PLN (*Bers and Despa, 2009*). PKA mediates phosphorylation of PLN at Ser16, which relieves its inhibition of SERCA, increasing muscle contractility and relaxation rate (*Tada et al., 1976*; *Simmerman et al., 1986*; *Zhao et al., 2004*; *Ablorh et al., 2014*).

To date DCM mutations associated with PLN include R9C, R9H, R9L, ΔR14, R14I, and I18T (*Schmitt et al., 2003*; *Medeiros et al., 2011*; *Haghighi et al., 2006*; *Burns et al., 2017*; *Schmitt et al., 2009*; *DeWitt et al., 2006*). These mutations cluster in a small 'hotspot' region that has little direct contribution to the inhibition of SERCA activity (*Kimura et al., 1996*). However, its neighborhood contains the critical PKA (*Tada et al., 1976*; *Simmerman et al., 1986*; *Ablorh et al., 2014*) and CaMKII (*Zhao et al., 2004*; *Mattiazzi et al., 2005*) phosphorylation sites, Ser16 and Thr17, respectively, suggesting a connection between altered regulation of phosphorylation and DCM phenotype. Among them, the R9C and ΔR14 mutations have the highest frequency and are associated with the most severe DCM phenotype (*Schmitt et al., 2003*; *Haghighi et al., 2006*). R9C PLN has been studied extensively, but its DCM-causing molecular mechanism remains controversial. The initial study by Schmitt et al. suggests that compared to wild-type (WT) PLN, the R9C mutant interacts more tightly with PKA, prevents dissociation of PKA from R9C PLN, and thus locks it in an inactive state, which prevents the phosphorylation of PLN at Ser16 (*Schmitt et al., 2003*). Ha et al. show that the R9C mutation stabilizes the pentameric form of PLN by introducing intersubunit disulfide bonds under oxidizing conditions, decreasing inhibition of SERCA (*Ha et al., 2011*). Other hypotheses highlight the importance of altered PLN conformation (*Yu and Lorigan, 2014*), hydrophobicity (*Ceholski et al., 2012b*), and PLN–membrane interactions (*Yu and Lorigan, 2013*) caused by the mutation. The impacts of other DCM mutations have been confirmed by genetic (*Burns et al., 2017*) or animal studies (*Haghighi et al., 2006*), but their disease mechanisms are far from clear.

ALN is a newly identified protein which possesses a transmembrane domain that shares the conserved SERCA-interacting sequence motif with PLN (*Anderson et al., 2016*). Unlike PLN, which is specifically expressed in cardiac muscle, ALN is ubiquitously expressed in many tissues including atria and ventricle but with the highest expression levels in the ovary and testis (*Anderson et al., 2016*). Although PLN and ALN diverge significantly in their cytoplasmic domains, a putative PKA recognition motif is present in ALN, suggesting the possibility that ALN is a substrate of PKA. Consistent with this notion, phosphorylation of Ser19 was detected by mass spectrometry of mouse ALN extracted from

various organs (*Huttlin et al., 2010*; *Lundby et al., 2013*). However, it remains to be investigated whether this phosphorylation is carried out by PKA and whether this phosphorylation regulates the interaction of ALN with SERCA.

Here, we report three crystal structures of the PKA catalytic domain (PKAc) in complex with three peptide variants of PLN, WT, R9C, and A11E. The long sought after structure of the PKAc–R9C PLN complex is critical for understanding the disease mechanism of familial DCM. Compared to the PKAc-WT PLN structure, the replacement of Arg by Cys abolishes an important electrostatic interaction, resulting in a significant conformational change of PLN and significantly reduced interactions between the two proteins. The binding affinities of various PLN peptides to PKAc were measured by surface plasmon resonance (SPR) and compared to each other. Consistent with our crystal structures, upon R9C mutation, the binding affinity to PKAc was significantly reduced compared to WT PLN. The kinetic constants of PKAc-catalyzed phosphorylation of PLN peptides were also determined and a significantly lower $k_{cat}/K_M$ was observed for R9C PLN. Our data also support the idea that other PLN mutations in the neighborhood, including DCM-related mutations at the 9th, 14th, and 18th positions, share a common disease mechanism related to reduced PKA phosphorylation. The solution-phase structures of free PLN variants determined by nuclear magnetic resonance (NMR) show that, individually, phosphorylation of the WT peptide and DCM-related mutations cause PLN to become more rigid, which might also contribute to the reduction of phosphorylation. We also confirm that ALN can be phosphorylated by PKA but not as efficiently as PLN. In addition, surprisingly, major differences between a previously published PKAc-WT PLN structure (PDB ID 3O7L) and ours were observed. Our structural models for the three complexes all show a monomeric form of PKAc that forms a 1:1 complex with PLN, which is consistent with the solution behavior of PKAc. In contrast, the 3O7L structure shows PKAc forming a dimer that makes a 2:1 PKAc:PLN complex in the crystal. Further analysis of the 3O7L structure indicates that the dimeric assembly of PKAc is likely an artifact due to crystal packing. The validity of our model is further supported by biophysical and biochemical assays with a series of PLN mutants, designed based on the observed interactions between PLN and PKAc in our structure. Thus, our structure represents only the second physiologically relevant structure of a PKAc–substrate complex, besides the PKAc–ryanodine receptor 2 (RyR2) complex (*Haji-Ghassemi et al., 2019*).

## Results

### Structures of PKAc in complex with WT PLN and PLN R9C

R9C is the most well-known PLN mutation associated with DCM. Despite extensive functional studies, the structural basis of this disease-causing mutation remains elusive. Here we present the crystal structures of PKAc in complex with a peptide corresponding to residues 8–22 of WT PLN, at 2.1 Å resolution (*Table 1*, *Figure 1A*), and the R9C variant of this PLN peptide, at 3.4 Å resolution (*Table 1*, *Figure 1B*). Adenosine 5′-(β, γ-imido) triphosphate (AMP-PNP), a nonhydrolyzable analog of the ATP cosubstrate, is also bound to PKAc in both structures. The peptides contain the phosphorylation site. The WT PLN peptide has previously been shown to be a good model substrate that gets phosphorylated as efficiently as the full-length PLN protein (*Masterson et al., 2010*; *Ceholski et al., 2012a*). The electron densities for the majority of the peptides (corresponding to PLN residues 8–19) and AMP-PNP are well defined in the structures (*Figure 4—figure supplement 1*, *Figure 1—figure supplement 1*). Two $Mg^{2+}$ atoms are observed in the catalytic site, similar to other reported PKA structures (*Kovalevsky et al., 2012*). In both cases, PKAc crystallized in a closed conformation with PLN docked to the large lobe and AMP-PNP bound with the small lobe.

In the complex of PKAc with WT PLN, the peptide substrate adopts an extended conformation. The N-terminal region (NTR) of WT PLN (Thr8 to Ile12) makes extensive interactions with helix 4 of the large lobe of PKAc. The binding is mainly mediated by an electrostatic interaction between the positively charged side chain of PLN Arg9 and the negative dipole moment of α-helix 4 and by a hydrophobic interaction between Ala11 of PLN and Phe129 of PKAc (*Figure 1C*). The hydroxyl group of the phosphorylation site, Ser16, is ~3.4 Å away from the γ-phosphate group of AMP-PNP, similar to other reported PKA structures (*Kovalevsky et al., 2012*). The asymmetric unit (ASU) of our structure contains one PKAc bound to one PLN (*Figure 1A*).

**Table 1.** Data collection and refinement statistics for the PKAc–phospholamban (PLN) crystals.

| Crystal | PKAc-WT PLN | PKAc-A11E PLN | PKAc-R9C PLN |
|---|---|---|---|
| $\lambda$ for data collection (Å) | 0.9795 | 0.9795 | 0.9795 |
| **Data collection** | | | |
| Space group | P 2 21 21 | P 1 21 1 | C 2 2 21 |
| *Cell dimension (Å)* | | | |
| a, b, c (Å) | 52.57, 70.49, 99.03 | 49.56, 69.37, 56.24 | 50.91, 104.88, 168.10 |
| $\alpha$, $\beta$, $\gamma$ (°) | 90.00, 90.00, 90.00 | 90.00, 101.97, 90.00 | 90.00, 90.00, 90.00 |
| Resolution | 42.14–2.16 | 33.12–2.80 | 44.19–3.43 |
| Rmerge | 0.194 (0.907) | 0.139 (0.527) | 0.165 (0.738) |
| Average $I/\sigma(I)$ | 9.9 (1.8) | 8.5 (1.8) | 10.3 (2.5) |
| Completeness (%) | 91.42 (68.22) | 99.11 (94.08) | 96.35 (95.02) |
| Redundancy | 6.9 (6.1) | 3.3 (3.1) | 5.4 (5.7) |
| **Refinement** | | | |
| Resolution (Å) | 42.14–2.16 | 33.12–2.80 | 44.19–3.43 |
| Highest resolution shells (Å) | 2.24 (2.16) | 2.78 (2.83) | 3.56 (3.43) |
| No. of reflections | 18,585 | 9242 | 6088 |
| Average $B$-factor | 30.88 | 37.93 | 95.81 |
| Protein | 30.52 | 37.97 | 95.92 |
| Ligands | 24.40 | 34.94 | 86.85 |
| Water | 36.01 | 37.30 | – |
| $R_{\text{work}}$ | 0.179 (0.230) | 0.203 (0.294) | 0.257 (0.339) |
| $R_{\text{free}}$ | 0.227 (0.290) | 0.252 (0.415) | 0.317 (0.360) |
| RMSD length (Å) | 0.008 | 0.002 | 0.001 |
| RMSD angle (°) | 1.20 | 0.490 | 0.370 |
| **No. of atoms** | | | |
| Protein | 2780 | 2672 | 2775 |
| Ligands | 33 | 33 | 33 |
| Water | 238 | 23 | 0 |
| **Ramachandran plot (%)** | | | |
| Most favored | 95.85 | 95.18 | 94.07 |
| Additionally allowed | 4.15 | 4.82 | 5.64 |

In the structure of the PKAc complex with PLN R9C, the NTR diverges from the structure of WT PLN bound to the enzyme. The mutation to Cys abolishes the interaction between the positively charged side chain of Arg9 and the negatively charged helix dipole at the C-terminal end of helix 4 from PKAc and a hydrogen-bond between PLN Arg9 and the main chain of PKAc Arg134, which releases the NTR of PLN from binding to the large lobe (*Figure 1C, D*). R9C turns the NTR counter-clockwise by 26° around the hinge formed by Ala11 and shifts the C$_\beta$ of residue Thr8 by 5.2 Å. The surface area of the enzyme–peptide interface decreases by 62.9 Å$^2$, as a consequence of the R9C mutation, which is predicted to strongly diminish the binding of PLN. The structural change in the NTR allosterically affects the conformation of the catalytic center. The C$_\alpha$ of Ser16 is shifted by 0.7 Å, which results in a conformational change of the side chain of Ser16, a displacement of AMP-PNP, and

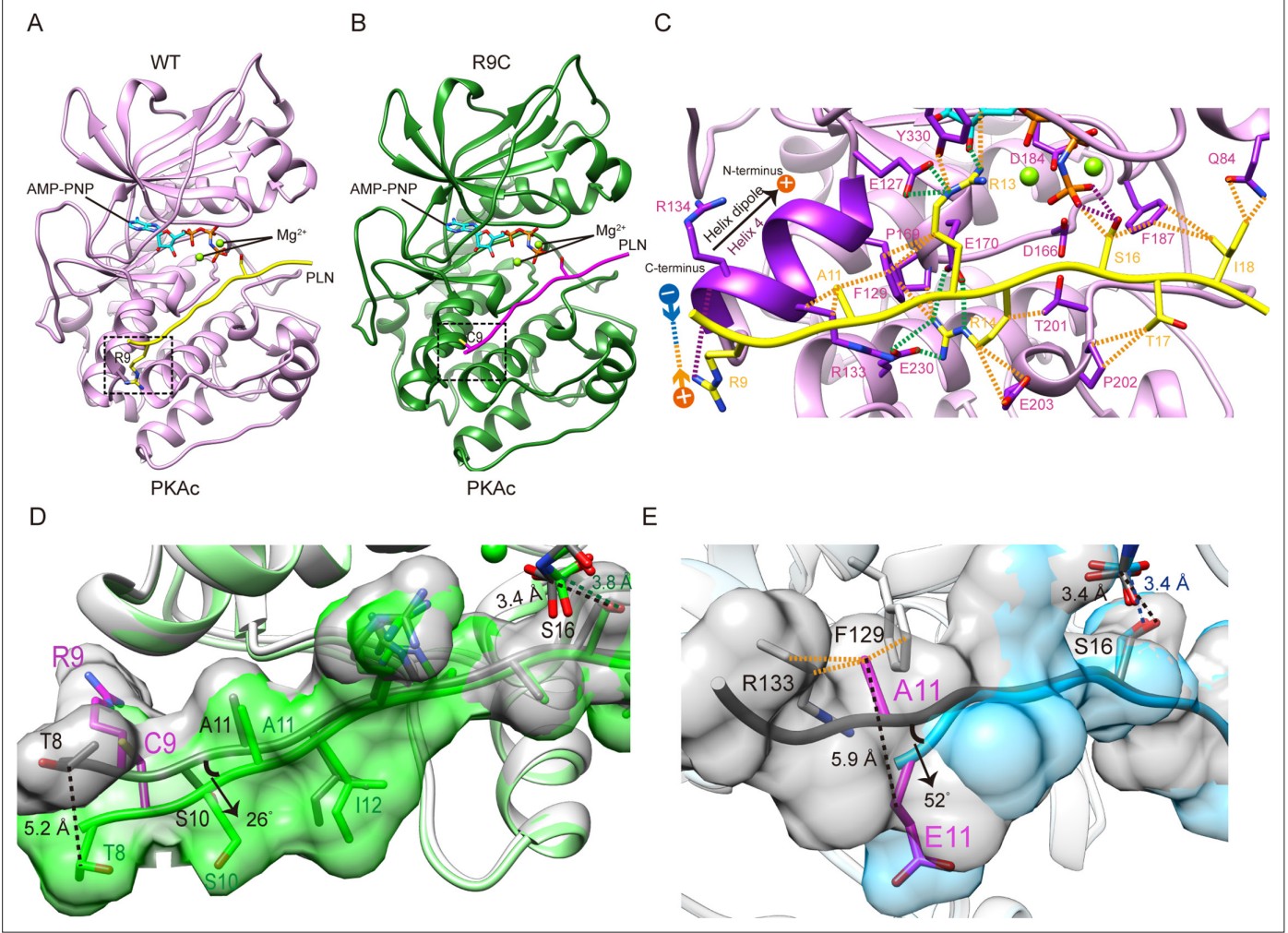

**Figure 1.** Crystal structures of PKAc-WT/R9C phospholamban (PLN) complex. (**A**) Crystal structures of the ternary complex of PKAc, WT PLN, and AMP-PNP. Protein kinase A (PKA) is colored in pink, PLN in yellow, and AMP-PNP in cyan. (**B**) Crystal structures of the ternary complex of PKAc, R9C PLN, and AMP-PNP. PKA is colored in green, PLN in violet red, and AMP-PNP in cyan. (**C**) The interaction between PKAc and WT PLN. The van der Waals contacts (orange), the salt bridges (green), and the hydrogen bonds (purple) are indicated by the dash lines. (**D**) The superposition of PKAc-WT PLN (white-gray) with PKAc-R9C PLN (light green-green). R9C abolishes the electrostatic interaction between Arg9 and the helix dipole of helix 4, inducing conformational changes at the N-terminal region (NTR). (**E**) The superposition of PKAc:WT PLN (white-gray) with PKAc:PLN A11E (light blue-cyan). A11E forces the NTR to move away from PKAc without affecting the structure at the catalytic center.

The online version of this article includes the following figure supplement(s) for figure 1:

**Figure supplement 1.** Electron density maps.

**Figure supplement 2.** Nuclear magnetic resonance (NMR) structures of phospholamban (PLN)/ALN peptides in the absence of PKAc.

**Figure supplement 3.** Sequence alignment between phospholamban (PLN) and ALN.

**Figure supplement 4.** Schematic representation showing the impacts of dilated cardiomyopathy (DCM) mutations and calmodulin-dependent protein kinase II (CaMKII) phosphorylation on the phosphorylation of phospholamban (PLN) by protein kinase A (PKA).

a subsequent ~0.4 Å increase in the distance between the γ-phosphate of AMP-PNP and the hydroxyl group of Ser16 (**Figure 1D**).

## DCM mutations reduce the binding of PLN and activity of PKA

To test whether the DCM mutations affect the binding between PKAc and the PLN-derived peptide, we characterized their interactions by SPR. To understand the interactions between the three components in the PKAc/AMP-PNP/substrate ternary complex, we first measured the affinity between PKAc and AMP-PNP in the absence of substrate. It shows a dissociation constant ($K_D$) around 110 μM

(*Figure 2—figure supplement 1A*). The binding affinity of WT PLN peptide to PKAc is clearly influenced by AMP-PNP: in the presence of 500 μM AMP-PNP the $K_D$ is ~180 μM (*Figure 2A*, *Figure 2—figure supplement 1C*), similar to the reported affinities for other PKA substrates such as kemptide and ryanodine receptor (RyR) (*Haji-Ghassemi et al., 2019*; *Masterson et al., 2008*), while in the absence of nucleotide, no binding could be detected (*Figure 2—figure supplement 1B*). Thus, we chose to include 1 mM of AMP-PNP for all the following SPR experiments involving the formation of ternary complexes.

The $K_D$ value of PLN R9C is about fourfold higher than that of WT PLN (*Figure 2A, B*). This difference confirms that the loss of interactions between the NTR of the peptide and the large lobe of PKAc is linked to a decrease in affinity for PLN R9C. We further tested the impact of four other DCM mutations, including R9H, R9L, ΔR14, and I18T, as well as an artificial mutation A11E, on the interaction between PKAc and PLN. Generally, all of them decrease the binding affinity compared to the WT PLN (*Figure 2C–G*). Among all the DCM mutations, R9H, the least deleterious of the disease-associated mutations, is the mildest, with an affinity 3.2-fold lower compared to WT. The replacement by histidine partially retains the positive charge at this position and might keep weak contact with the negatively charged helix dipole of helix 4 (*Figure 1C*). The R9C PLN peptide, which should lack any positive charge character at this position, has a slightly higher $K_D$ value compared to the R9H variant. In contrast, the replacement of Arg9 by leucine, which has a nonpolar side chain, shows a much larger weakening effect. Arg14, from the classic R-R-X-S/T motif, forms extensive interactions with PKAc, involving a salt bridge network with Glu170 and Glu230 and van der Waals contacts with Phe129, Thr201, Pro169, and Glu203 of PKAc (*Figure 1C*), similar to what was seen in previous studies with other peptides known to bind PKAc (*Kovalevsky et al., 2012*; *Bossemeyer et al., 1993*; *Breitenlechner et al., 2005*; *Breitenlechner et al., 2004*; *Lauber et al., 2016*; *Pflug et al., 2012*; *Engh et al., 1996*; *Rouse et al., 2009*). Therefore, it is not surprising that the deletion of Arg14 can cause a dramatic tenfold reduction in binding affinity because it will not only cause the change at Arg14, but also make all residues upstream of residue 14 out of register. The I18T mutation has a similar effect on the affinity as the deletion of Arg14. Ile18 forms extensive van der Waals contacts with Gln84 and Phe187 of PKAc. Replacement by the smaller and more hydrophilic threonine would cause the loss of contacts, which weakens the binding (*Figure 1C*). PLN A11E exhibits an approximately twofold elevated $K_D$ value, relative to WT PLN. Given the increases in side-chain size and polarity, this mutation likely disrupts the interaction between the methyl group of Ala with Phe129 on α-helix 4 of PKAc, causing the affinity to decrease.

The strength of the interactions between PKAc and substrate peptides was also examined by measuring the thermal stability of the complexes in the solution phase. The addition of WT PLN peptide to PKAc increases its melting temperature ($T_m$) by 1.4°C. In contrast, the mutant PLNs show less contribution to the increase of PKAc thermal stability (*Figure 3A, B*). Among the five mutations, R9L and ΔR14 show the least stabilizing effects, consistent with the SPR result. Thus, the observed affinity decreases of the DCM-associated peptide variants are not an artifact of PKAc immobilization. Considering the peptides were given in excess, the results of the thermal shift assay does not reflect the quantitative percentage binding but rather the difference in the interaction mode between the peptides and PKA.

The enzyme kinetic constants of PKAc for WT- and R9C-PLN peptides were determined by an ADP-Glo kinase assay (*Figure 3C*). The turnover numbers ($k_{cat}$) are generally in the same ballpark with previously reported $k_{cat}$ values determined using PLN, kemptide, SP20, and RyR2, as substrates (*Haji-Ghassemi et al., 2019*; *Masterson et al., 2010*; *Trafny et al., 1994*; *Mitchell et al., 1995*). The $k_{cat}$ values for both PLN variants remain relatively unchanged, suggesting that the increased distance and altered orientation between the hydroxyl group of Ser16 in PLN R9C and the gamma-phosphate of AMP-PNP do not significantly impact transition state stabilization during catalysis of phosphate group transfer. The $K_M$ value for PLN R9C is twofold higher than WT PLN (*Table 2*, *Figure 3C*). This difference in $K_M$ and $K_D$ values is similar for these peptide substrates, which could indicate that the lower catalytic efficiency seen with PLN R9C is mostly due to decreased substrate binding to the enzyme. Our observations suggest that the phosphorylation level of R9C PLN should be lower compared to WT PLN under physiological conditions, which is consistent with previous measurements of their phosphorylation levels in cells made by western blot (*Ha et al., 2011*; *Schmitt et al., 2003*; *Kim et al., 2015*). Next we performed ADP-Glo assays with the other DCM-associated peptide variants, as well

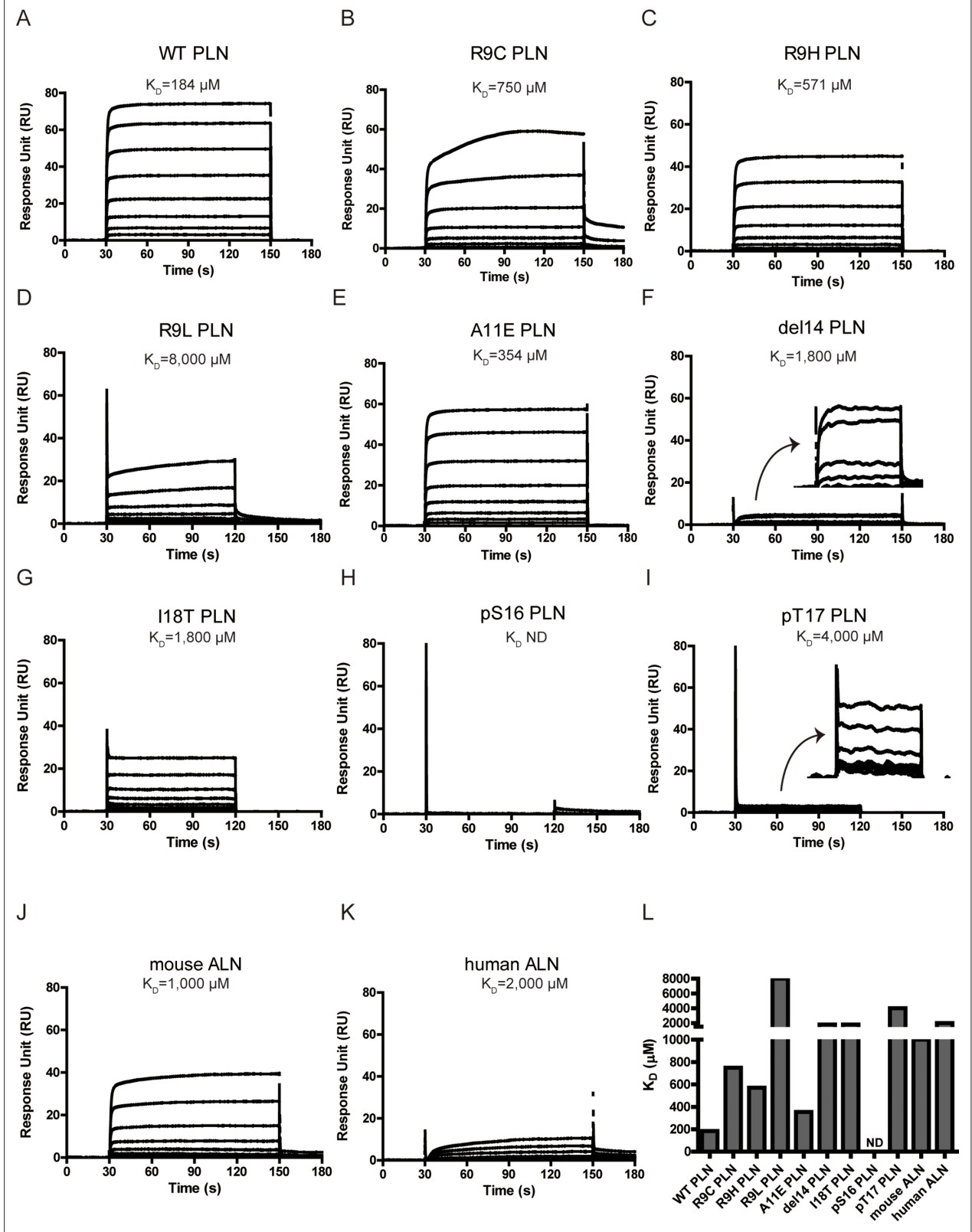

**Figure 2.** Surface plasmon resonance (SPR) analysis of PKAc–phospholamban (PLN)/ALN interactions. (**A–K**) SPR sensorgrams of the binding of the PLN/ALN peptides onto immobilized PKAc. The calculated $K_D$ values are displayed above the corresponding sensorgrams. (**L**) The relative $K_D$ values of PKAc with different peptide substrates measured by SPR assay.

*Figure 2 continued on next page*

*Figure 2 continued*

The online version of this article includes the following figure supplement(s) for figure 2:

**Figure supplement 1.** The surface plasmon resonance (SPR) analysis of PKAc/AMP-PNP interactions.

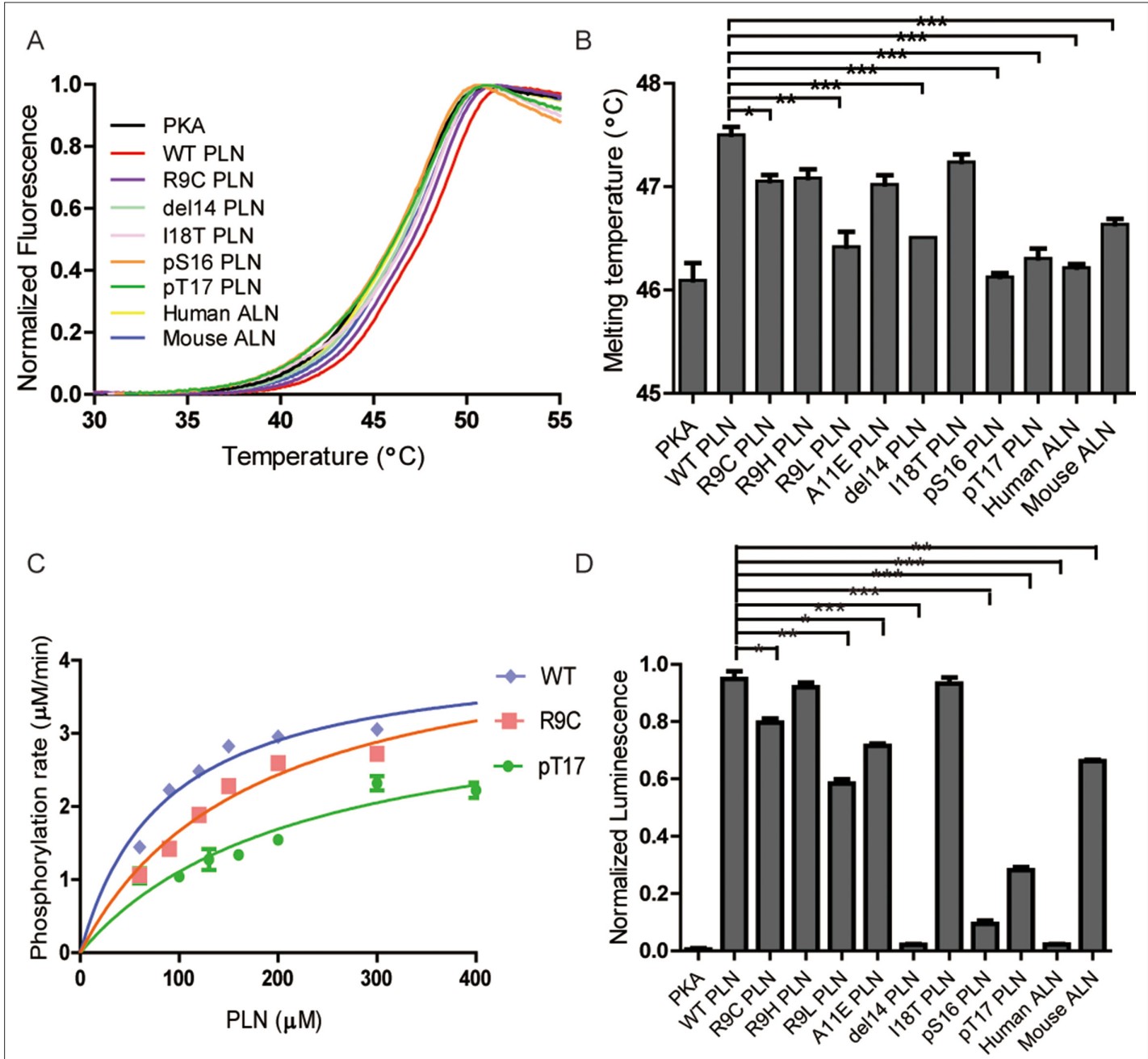

**Figure 3.** Thermal melt analysis of PKAc–phospholamban (PLN)/ALN complexes and the activities of PKAc measured by ADP-Glo assay. (**A**) The averaged thermal melt curves from four replicates each of PKAc complexed with different peptides. (**B**) A bar graph comparing the melting temperatures of PKAc complexed with different peptides. Error bars show the standard deviation. ***p < 0.0001, **p < 0.001, *p < 0.01 (one-tail Student's *t*-test). (**C**) A plot of vi vs. [PLN peptide] for PKAc-catalyzed phosphorylation of WT-, R9C-, and pT17-PLN substrates. The data were fit to the Michaelis–Menten equation. (**D**) The relative activities of PKAc with different peptide substrates measured by ADP-Glo assay.

**Table 2.** Enzyme kinetic parameters for PKAc-catalyzed phosphorylation of different phospholamban (PLN) substrates measured by ADP-Glo assay.

|  | WT PLN | PLN R9C | pThr17 PLN |
|---|---|---|---|
| $V_{max}$ (µM/min) | 4.1 ± 0.23 | 4.5 ± 0.33 | 3.6 ± 0.46 |
| $K_M$ (µM) | 85 ± 13 | 173 ± 25 | 223 ± 57 |
| $k_{cat}$ (s$^{-1}$) | 6.9 ± 0.38 | 7.6 ± 0.6 | 6.0 ± 0.7 |
| $k_{cat}/K_M$ (s$^{-1}$ M$^{-1}$) | $8.1 \times 10^4$ | $4.4 \times 10^4$ | $2.7 \times 10^4$ |

as PLN A11E, as substrates at a given concentration near the $K_M$ value of WT PLN. Among all mutants tested, R9H has the mildest effect, while ΔR14 shows the largest decrease of PKAc catalytic efficiency (*Figure 3D*). The enzyme activities of PKAc for different substrates show a roughly similar pattern with their binding affinities determined by our biophysical assays (*Figure 3B, D*), highlighting the importance of the binding affinity of PLN to PKA in DCM disease models.

## Comparison of PKAc–PLN structure with a previously reported crystallographic model

Surprisingly, we find that our structure exhibits substantial differences compared to the previously published structure of the complex between PKAc and a peptide corresponding to the first 19 amino acids of human PLN complex structure (PDB ID 3O7L) (*Figure 4A–C*; *Masterson et al., 2010*). The overall root mean square deviation (RMSD) between 3O7L and our structure of the complex between PKAc and the WT PLN peptide (corresponding to amino acids 8–22 of human PLN) is only ~0.6 Å, but the RMSD between all modeled Cα atoms of the PLN portions is over 4.4 Å (*Figure 4—figure supplement 2*). Another significant difference is that the ASU of our structure contains only one PKAc bound to one WT PLN peptide (*Figure 4A*). In contrast, the ASU of 3O7L contains two PKAc molecules and one bound PLN whose NTR shows a substantially different conformation and interacts with both PKAc molecules. The second PKAc (Mol B) from the ASU is in a closed noncatalytic conformation. Nonetheless, PKAc Mol B makes extensive contacts with the PLN ligand, particularly with the side chains of Tyr6, Leu7, Thr8, and Ser10 (*Figure 4B*). An interface area calculation of 3O7L shows that 29% of the interactions between PLN and PKAc originate from Mol B. In contrast, the interactions of PLN with PKAc in our structure originate mainly from a single PKAc molecule within the same ASU (*Figure 4A*).

In order to examine the quaternary structure in solution, purified PKAc complexes were subjected to analytical size-exclusion chromatography. PKAc elutes as expected for a monomer with or without PLN peptide in the presence of AMP-PNP (*Figure 4—figure supplement 3A–C*). The monomeric assembly of PKAc:PLN peptide complexes is unaffected by N-terminal truncation, as peptide ligands corresponding to amino acids 1–19 and 8–19 of human PLN eluted similarly. Likewise, the R9C mutation in the PLN sequence did not change the assembly state of the enzyme:peptide complex (*Figure 4—figure supplement 3D, E*). Therefore, the interactions that PLN makes with Mol B in the 3O7L structure do not exist in solution but only occur due to the crystal packing. In all other available complex structures of PKAc, there is also only one PKAc molecule bound with one substrate or inhibitor, suggesting the 1:1 ratio should be the common physiological form (*Kovalevsky et al., 2012*; *Bossemeyer et al., 1993*; *Breitenlechner et al., 2005*; *Breitenlechner et al., 2004*; *Lauber et al., 2016*; *Pflug et al., 2012*; *Engh et al., 1996*; *Rouse et al., 2009*; *Oebbeke et al., 2021*).

A further difference lies in the active site. Our model shows clear electron density for AMP-PNP and the nearby glycine-rich loop (gly-loop) (*Figure 4—figure supplement 1*, *Figure 1—figure supplement 1*). In 3O7L, the γ-phosphate of AMP-PNP was not modeled, and there is a clear negative difference density for the rest of molecule according to the map generated using the previously deposited structure factor data, reflecting a low occupancy of the nucleotide (*Figure 4D*, *Figure 4—figure supplement 1*). The re-refined coordinates and map in the PDB-REDO database are improved compared to the original 3O7L, showing that the nucleotide and first two phosphates are present while the γ-phosphate is likely hydrolyzed (*Figure 4—figure supplement 1*). Further, the 3O7L structural model contains a PEG molecule that is located in a patch of negative difference density, which raises the question of whether or not it is actually present. Neighboring positive difference density is more likely to correspond to the gly-loop according to the comparison of the two structures (*Figure 4C, D*,

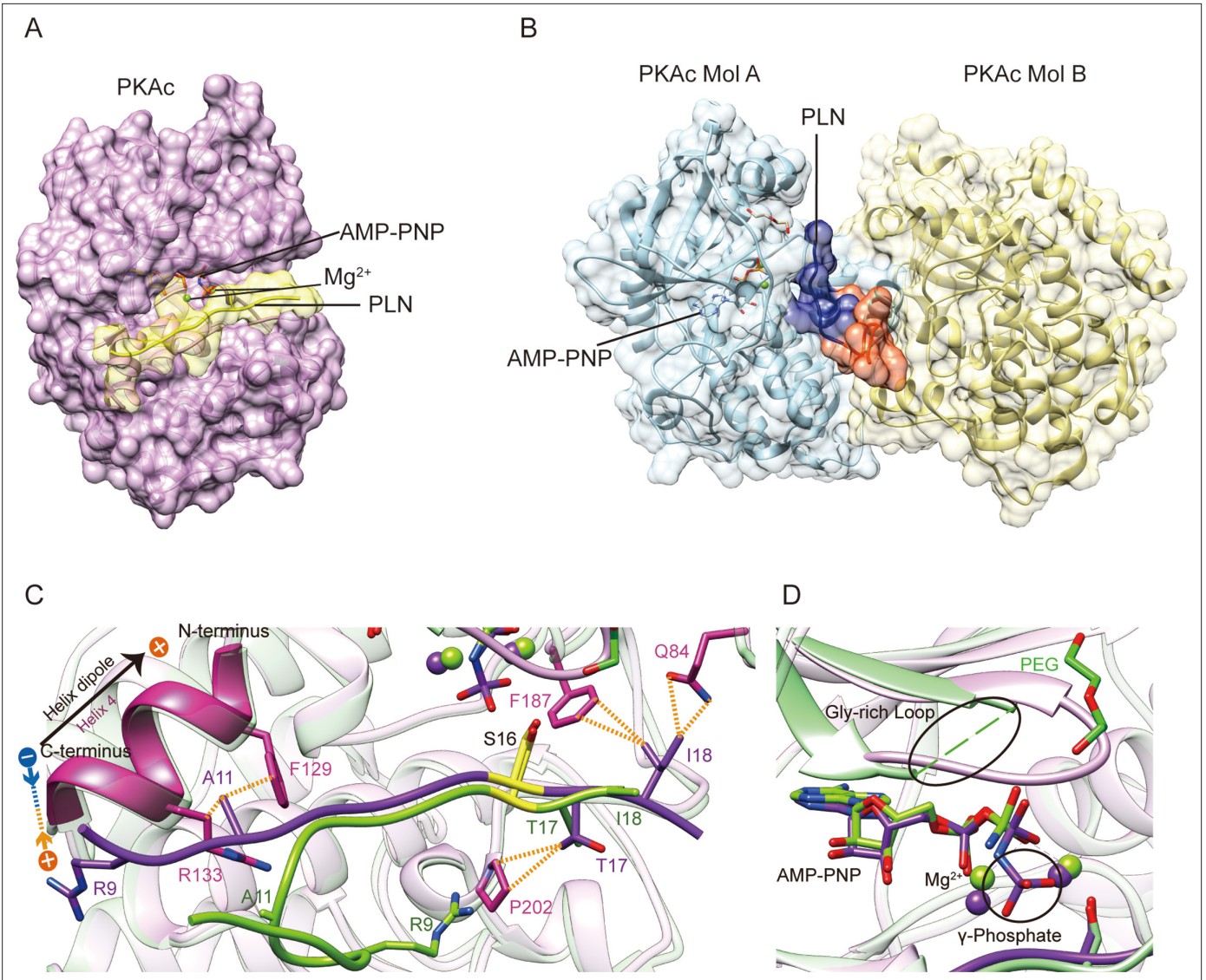

**Figure 4.** Crystal structures of PKAc–phospholamban (PLN) complex. Crystal structures of the ternary complex of PKAc, PLN, and AMP-PNP from the current study (**A**) and a previous study (PDB ID 3O7L)[58] (**B**). PLN interacts with a single PKAc in our structure and with two PKAc molecules in 3O7L. The N-terminal region (NTR) of PLN interacts extensively with the second PKAc (Mol B) in 3O7L. (**C, D**) Superposition of our structure (purple) and 3O7L (green) shows that PLNs adopt different conformation at both ends. The γ-phosphate group in AMP-PNP and Gly-rich Loop is missing from 3O7L. The electrostatic interaction between Arg9 and the helix dipole is indicated. The PKA phosphorylation sites are highlighted in yellow.

The online version of this article includes the following figure supplement(s) for figure 4:

**Figure supplement 1.** Comparison of two structures of PKAc–phospholamban (PLN) complex.

**Figure supplement 2.** Root mean square deviation (RMSD) plots.

**Figure supplement 3.** Oligomerization states of PKAc.

*Figure 4—figure supplement 1*). The gly-loop is right next to the nucleotide and is crucial for nucleotide coordination. Thus, it would be very unusual for a PEG molecule to occupy this loop position, especially when the nearby catalytic loop residues Asp166 and Lys168 are in similar positions in both structures.

The last difference is in the C-terminal region of PLN. One of the DCM mutation sites, Ile18 of PLN, shows extensive interactions with Gln84 and Phe187 of PKAc in our model; in contrast, it is mainly facing solvent in 3O7L (*Figure 1C*). The results from our functional assays (see above) show that the disease mutation I18T clearly reduces the phosphorylation level of PLN and its binding affinity with PKAc, which agrees with our structural model.

## Structure of PKAc in complex with DCM PLN-mutant A11E

Residue Ala11 forms hydrophobic interactions with the side chain of Phe129 and the β and γ carbons of Arg133 in our PKAc:WT PLN complex structure; in contrast, the same residue is solvent exposed and is not involved in any interaction with PKAc in 3O7L (*Figure 1C*). Therefore, to distinguish whether A11E mutation forms contact with PKA or not, we solved the structure of PKAc in complex with the A11E PLN peptide at 2.8 Å resolution (*Table 1*, *Figure 1E*). This complex confirms that the mutant PLN can still bind PKAc but with fewer interactions, which explains the decrease in affinity. The mutation flips the side chain of residue 11 and pushes the NTR of PLN away from the large lobe of PKAc (*Figure 1E*). The $C_\beta$ of Glu11 moves 5.9 Å away from PKAc compared to the $C_\beta$ of Ala11, but the structures of their catalytic centers, including the catalytic loop, gly-loop, two $Mg^{2+}$, and AMP-PNP, are similar. Together with the clear functional effect of A11E (*Figures 2E, 3B, D*), we propose that Ala11 contributes to the binding of PLN to PKAc as seen in our crystal structure of the PKAc:WT PLN complex.

## Phosphorylation at Ser16 and Thr17 reduce the binding of PLN and activity of PKA

Next, we tested whether phosphorylation at Ser16 (PKA site) and Thr17 (CaMKII site) would affect the binding of PLN with PKAc. The Ser16 side chain points directly to the catalytic center of PKAc. Thus, its phosphorylation would cause steric hindrance with Phe187 and charge repulsion with Asp166, Asp184, and also the gamma-phosphate group from ATP (*Figure 1C*). This is supported by the previous observation that the $K_D$ value of a phosphoserine containing peptide product of PKAc is increased by ~170-fold compared to the nonphosphorylated substrate (*Granot et al., 1981*). As expected, we could not detect any significant binding between PLN pSer16 and PKAc by SPR (*Figure 2H*). In comparison, the Thr17 side chain interacts with the side chain of Pro202 (*Figure 1C*), so we predict that its phosphorylation would also reduce the binding to PKAc, but to a lesser extent. Indeed, PKAc shows a 20-fold weaker binding toward the phosphorylated Thr17 peptide substrate, but still detectable by SPR, with a $K_D$ ~4 mM (*Figure 2I*). pSer16 shows a $T_m$ value similar to the negative control (PKAc in the absence of PLN), while pThr17 shows a slightly higher $T_m$ value (*Figure 3A, B*), which confirms their low affinities for PKAc found by SPR. The relative activity of PKAc on pThr17 is less than 1/3 of WT PLN, which is mostly due to an increased $K_M$ value for this substrate. Only a small residual activity was observed for pSer16, probably due to a small percentage of hydrolyzed pSer16 PLN substrate (*Figure 3D*). The kinetic behaviors of these substrate variants thus reflect their decreased affinities for PKAc.

## Structural dynamics determined by NMR

To find out how phosphorylation and sequence variations affect the conformation of PLN in the absence of PKA, we solved the structures of peptides corresponding to segments of WT PLN, R9C PLN, pSer16 PLN, and pThr17 PLN by NMR. We analyzed the 20 lowest energy conformations from all four peptide variants. The WT PLN peptide clearly shows a more dynamic conformation whose structures can be classified into five distinct conformations using Chimera Ensemble Cluster (*Pettersen et al., 2004*; *Figure 1—figure supplement 2A*). The dynamic nature of WT PLN can be reflected by the relatively high RMSD value calculated by comparing the representative structures from each ensemble (*Figure 1—figure supplement 2B*). In comparison, the R9C, pSer16, and pThr17 PLN variants show a relatively low RMSD among the 20 lowest energy conformations, indicating that these peptides are all less flexible than WT PLN. The structural differences between the R9C-, pSer16-, pThr17-, and WT-PLN peptides might be related to the local charge changes induced by the mutation or phosphorylation, which further affect the intramolecular electrostatic interactions with positively charged Arg13 and Arg14 (*Figure 1—figure supplement 2C*). The lower flexibility of R9C PLN and pThr17 PLN might further help to explain their decreased ability to bind PKAc. While none of the conformations of the four peptide variants seem to be significantly preorganized for binding to the PKA active site, we propose that it might take less energy to rearrange/restructure WT PLN to a proper 'bound conformation' before it can be phosphorylated by PKA. If so, both indirect (more energetically costly conformational rearrangement of the peptide during enzyme binding) and direct (loss of a stabilizing electrostatic interaction with the enzyme) effects might contribute to the lower binding affinity (higher $K_D$ value) and less efficient conversion to product (higher $K_M$ value) of R9C PLN.

## General binding determinant in SERCA-regulating peptides

To study whether other SERCA-regulating peptides can also be phosphorylated by PKA, we tested its activity with another recently identified peptide, called ALN, which is ubiquitously expressed in many tissues. 11AIRRAST17 in human PLN aligns with 14RERRGSF20 in mouse ALN (*Figure 1—figure supplement 3*), and both segments contain the R-R-X-S/T PKA recognition motif. As expected, mouse ALN also acts as a PKA substrate, however, PKAc shows about 5-fold lower binding affinity and 1.5-fold lower activity toward mouse ALN compared to human PLN (*Figures 2J and 3D*). Our PKAc:PLN complex structures show that Ala11 forms a hydrophobic interaction with PKAc, and the replacement of arginine in mouse ALN at this position would introduce charge repulsion with the double arginine at position 133 and 134 of PKAc (*Figure 1C*). The substitution of Thr17 by the bulky hydrophobic Phe20 in mouse ALN might further cause a clash and reduce the interaction (*Figure 1C*). We also used human ALN, which lacks the serine phosphorylation site, as a negative control. As predicted, no binding and phosphorylation activity could be detected (*Figures 2K, 3B, D*).

## Discussion

It is controversial how the mutations in PLN cause DCM. While it is clear that the phosphorylation of PLN by PKA can release its inhibition of SERCA, several models have been proposed to show that the DCM mutations in PLN might change this regulation in either a phosphorylation-dependent or -independent manner (*Ceholski et al., 2012a*; *Kim et al., 2015*; *Traaseth et al., 2009*; *Li et al., 2003*; *Nelson et al., 2018*; *De Simone et al., 2013*). Our data provide structural and functional confirmation that DCM mutations can reduce the binding of PLN substrate to PKA and subsequently its phosphorylation level.

Our work supports a model in which the mutations at positions 9, 14, and 18 of PLN share a common disease mechanism. The cytoplasmic domain of PLN binds with PKAc in a 1:1 ratio through extensive interactions from several key residues, including Arg9, Arg14, and Ile18. The mutations at these three positions have two main effects: (1) change the conformation of the substrate before binding to PKA, as shown by NMR structures; and (2) reduce the binding affinity with PKA, as shown by SPR, thermal melt and ADP-Glo assays, via disruption of enzyme–substrate interactions, as revealed by comparison of the crystal structures of PKAc complexed to WT- and R9C-PLN peptides (*Figure 1—figure supplement 4*). Previously it has been proposed that in heterozygous individuals, the aberrant interaction of mutant PLN with PKA may sequester PKA and prevent phosphorylation of WT PLN (*Schmitt et al., 2003*; *Young et al., 2015*). However, our results show that it is not likely that the DCM-mutant PLNs can sequester PKA since they interact even more weakly compared to WT PLN. For the structures of PKAc in complex with the two mutant PLNs (R9C and A11E), the positions of the substrates near the PKA catalytic center are relatively conserved, which explains why their turnover numbers are nearly unchanged relative to WT PLN. Generally, the loss of interactions near the mutation site causes a reduction in ground state affinity and an increase in the $K_M$ value, which would result in a decreased phosphorylation level of PLN. Lower phosphorylation levels of PLN in cardiac cells would lead to greater inhibition of SERCA, decreasing heart muscle contractility and relaxation rate. While catalytic efficiency of PKAc with PLN R9C only decreases by ~twofold, a corresponding change in phosphorylation level of PLN could be consistent with the relatively mild symptoms of DCM, and even a small increase in SERCA inhibition resulting from such a decrease in PLN phosphorylation would likely compound the $Ca^{2+}$ imbalance in the cell over repeated cycles of cardiac muscle contraction and relaxation. Additionally, the trend of the reduced phosphorylation by DCM mutations can be significantly affected by the oligomerization state of PLN. Ceholski et al. showed that R9C severely inhibits PKA phosphorylation in the context of full-length pentameric PLN, but has a much milder effect in the context of full-length monomeric PLN or an isolated tail peptide (*Ceholski et al., 2012a*). While decreased PLN phosphorylation is likely an important contributor to the physiological dysfunction associated with familial DCM, disease-causing mutations in PLN may have additional consequences, such as altered assembly state of PLN, phosphorylation of PLN by CaMKII, or changes in interactions of PLN with the lipid membrane. The influence of such factors on SERCA inhibition is unclear. In principle, they might further increase inhibition of SERCA and act in conjunction with lower PKA-mediated phosphorylation to manifest the disease symptoms. Conversely, it is possible that these factors could

decrease the inhibition of SERCA, partially compensating for the decreased phosphorylation level, and mitigating the symptoms.

That might further increase inhibition of SERCA and act in conjunction with lower PKA-mediated phosphorylation to manifest the disease symptoms or decrease the inhibition of SERCA and compensate the symptoms.

In addition to changing the interaction with PKAc, mutation or phosphorylation changes the conformational flexibility of free PLN, which might be another reason for the observed decrease in binding. Our NMR results show that R9C-, pSer16-, and pThr17-PLN are generally more rigid compared to WT PLN, probably due to the change in surface charge. Thus, it requires more energy input to reorganize them before binding to PKAc. Previous NMR studies using full-length PLN in the presence of detergents also demonstrated that phosphorylation could change the dynamics of PLN (*Vostrikov et al., 2013*; *Abu-Baker and Lorigan, 2006*). A previous study on the intracellular calcium-release channel RyR2 shows that a phosphomimetic at a CaMKII site induces a conformational change from loop to helix, and thus forms a more rigid structure (*Haji-Ghassemi et al., 2019*), similar to the changes in substrate flexibility observed here. However, in that case, the CaMKII site is at a position 7 residues upstream of the PKA site. Therefore, the formation of the new helix stabilizes the interaction with PKAc instead of weakening it, as happens with PLN, where the CaMKII site is right next to the PKA site. Subsequently, phosphorylation at the CaMKII site in RyR2 increases the affinity and activity of PKA (*Haji-Ghassemi et al., 2019*), while phosphorylation at the CaMKII site (Thr17) of PLN clearly reduces its ability to be phosphorylated by PKA (*Figure 1—figure supplement 4*). Crosstalk between PKA and CaMKII has been reported in a few different cases (*Haji-Ghassemi et al., 2019*; *Popescu et al., 2016*; *Poláková et al., 2015*). Phosphorylation by one kinase could either facilitate or hinder phosphorylation by a second kinase, and in this way, it connects the signaling networks at different nodes. For PLN, the pThr17-PLN was reported to have the strongest inhibition of SERCA, followed by the pSer16/pThr17 double phosphorylated PLN, while pSer16 had the weakest inhibitory activity (*Ablorh et al., 2014*). Thus, the activation of CaMKII on top of the PKA activation could decrease SERCA activity through two related pathways, the reduction of the phosphorylation level on PKA phosphorylation site Ser16 and the weakening of the inhibitory effect of PLN considering pSer16/pThr17 and pThr17 inhibit SERCA more effectively compared to pSer16. It remains to be tested whether phosphorylation by PKA at Ser16 or DCM mutations in PLN also weaken phosphorylation by CaMKII.

So far, three DCM mutations (R9C, R9L, and R9H), with different population frequencies, have been identified at the same position on PLN, making Arg9 a DCM mutation hotspot. According to our WT and R9C structures, the replacement of Arg with any neutral amino acid would abolish an important electrostatic interaction between the positively charged arginine and the negatively charged helix dipole and subsequently reduce the phosphorylation level of PLN. The effects of the mutations at the position 9 seem to be correlated to the polarity of the side chain. Histidine, which could be weakly positive, shows the mildest effect, while leucine, which is highly hydrophobic, almost completely abolishes the binding. The effect of cysteine is between the above two replacements. Indeed, it has been shown that R9C and R9L can abolish the inhibition of SERCA, while R9H is more similar to WT PLN (*Ceholski et al., 2012a*; *Young et al., 2015*). It requires further investigation to determine whether the clinical severity of these mutations correlates with the change of phosphorylation level.

The previously published crystal structure partially misguided attempts to understand how PKA regulates PLN. There are three clear discrepancies between the previous structure and our structure of the PKAc:WT PLN complex. First, the previous structure shows a sandwich conformation of the complex containing two PKAc and one WT PLN, with the NTR of PLN interacting extensively with the second PKAc molecule in the ASU. This binding mode is clearly a crystallization artifact since the PKAc:WT PLN complex has a monomeric form in solution. Second, the electron density map of 3O7L has poor quality in the regions of AMP-PNP and PEG. Indeed, the difference density is not compatible with a PEG molecule and the area is most likely occupied by the catalytically important gly-loop. Third, the side chain of DCM mutation site Ile18 was modeled in a truncated form (with only the β-carbon remaining) and in a solvent-facing orientation, which cannot explain the decrease in PKAc binding caused by the DCM mutation I18T. This modeling of the Ile18 side chain could be due to the weak electron density in the previous structure. Our structure shows a different conformation of the Ile18 side chain, which clearly interacts with PKAc. Thus, our explanation for the differences between the two structures is as follows: 3O7L presents a 2:1 (PKAc:PLN) complex structure, where the PLN

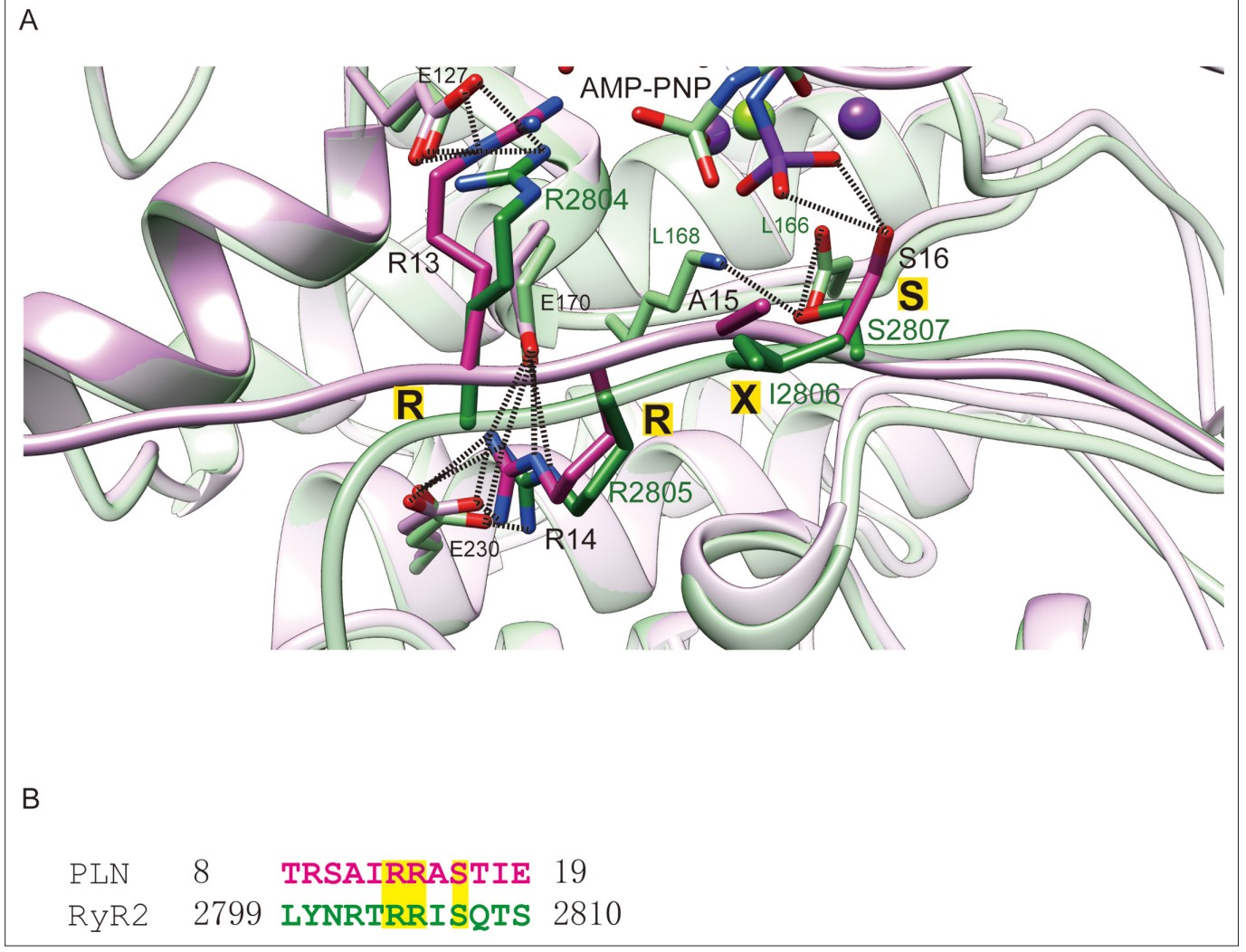

**Figure 5.** Comparison of PKAc–PLN and PKAc-RyR2. (**A**) Superposition of the crystal structures of PKAc–PLN (violet) and PKAc-RyR2 (green). (**B**) Sequence alignment of PKA-interacting fragments from PLN and RyR2. The conserved RRXS motif is highlighted.

peptide was trapped between two PKAc molecules from the same ASU, and the crystal contacts force the substrate into an unnatural pose that reduces the binding affinity of the nucleotide; our monomeric structure, which was generated using different crystallization conditions, presents a 1:1 (PKAc:PLN) complex structure, consistent with the native solution behavior and with a fully occupied nucleotide-binding site. These errors in the previous model would certainly compromise our understanding of the mechanism by which PKA regulates PLN. Our new structure of the PKAc:WT PLN complex shows clear electron densities in these key regions of the substrate-binding interface and the catalytic center, thus avoiding ambiguity in modeling and providing a more accurate structural template.

To identify general rules for PKA substrate binding, we compared the two available structures of PKAc in complex with their physiological substrates: RyR2 and PLN. As expected, the two PKAc molecules are similar to each other with an overall RMSD of 0.66 Å (*Figure 5A*). For the substrate, the classic PKA recognition motif 'RRXS' shows the highest structural similarity with an RMSD of 0.86 Å, while the NTRs of the substrates show greater divergence (*Figure 5A*). The interactions observed between Arg13 of PLN and Phe129, Glu170, Glu127, Tyr330 of PKA, and Arg14 of PLN and Glu170, Thr201, Glu203, Pro169, Glu230 of PKA, are conserved between several known PKA substrates (*Figure 5B*; *Kovalevsky et al., 2012*; *Bossemeyer et al., 1993*; *Breitenlechner et al., 2005*; *Breitenlechner et al., 2004*; *Lauber et al., 2016*; *Pflug et al., 2012*). This similarity confirms the importance of the RRXS motif in specific substrate recognition of PKA.

ALN is a newly identified SERCA-regulating peptide that is expressed more ubiquitously than PLN, particularly in the ovary and testis (*Anderson et al., 2016*). ALN has a longer cytoplasmic loop compared to PLN with a predicted PKA recognition motif (*Figure 1—figure supplement 3*). Its phosphorylation at Ser19 has been confirmed in liver, pancreas and heart tissue by mass spectrometry (*Huttlin et al., 2010*; *Lundby et al., 2013*), but the identity of the kinase remained unknown. Using in vitro PKA phosphorylation assays, we confirmed that ALN could indeed be phosphorylated by PKAc, although less efficiently compared to PLN. The physiological importance of this regulation remains to be investigated. The phosphorylation of ALN by PKA in mice but not humans could be relevant for understanding animal models of heart disease and how these animal models might behave differently from humans.

## Methods
### Cloning, expression, and purification of protein
The gene encoding mouse PKAc (gene accession number BC003238) was cloned into the pET-28a-HMT vector, which contains a hexahistidine tag, an MBP fusion protein and a TEV cleavage site at the N-terminus (*Lobo and Van Petegem, 2009*). For protein production, the plasmid was introduced into *Escherichia coli* BL21 (DE3) cells. Cells were grown at 37°C with shaking at 250 rpm in 2YT medium supplemented with 50 μg/ml kanamycin. When the $OD_{600}$ reached ~0.6, protein production was induced with 0.4 mM isopropyl-β-D thiogalactopyranoside (IPTG) and incubated at 18°C for another 24 hr. The cells were harvested by centrifugation at 8000 × *g* for 10 min and disrupted via sonication in lysis buffer (10 mM 10 mM N-2-hydroxyethylpiperazine-N-2-ethane sulfonic acid [HEPES], pH 7.4, 250 mM KCl, 10 mM 10 mM beta-mercaptoethanol [BME], 25 mg/ml DNase I, 25 mg/ml lysozyme, 1 mM phenylmethanesulfonyl fluoride [PMSF]). The cell debris was removed by centrifugation at 40,000 × *g* for 30 min. The soluble fraction was filtered through a 0.22 μm filter and loaded onto a 5 ml His Trap HP column (GE Healthcare) pre-equilibrated with buffer A (10 mM HEPES, pH 7.4, 250 mM KCl, 10 mM BME). The column was eluted using a linear gradient of 20–250 mM imidazole in buffer A. The eluted PKAc was cleaved with recombinant TEV protease at 4°C overnight, followed by purification using an amylose resin column (New England Biolabs) to remove the His-MBP-tag. The samples were loaded to an amylose column pre-equilibrated with buffer A, and eluted with the same buffer plus 10 mM maltose. The flow-through from the amylose column was loaded onto another HisTrap HP column (GE Healthcare) to further remove the fusion tag. PKAc was further purified using a SP Sepharose high-performance column (GE Healthcare) with a linear gradient from 20 to 500 mM KCl in elution buffer (10 mM Tris, pH 6.8, 10 mM BME). Finally, the PKAc was concentrated using Amicon concentrators (10 K MWCO from Millipore) and run over a Superdex 200 16/600 gel filtration column (GE Healthcare) in buffer A. The protein purity was examined by sodium dodecyl sulfate–polyacrylamide gel electrophoresis with a 15% (wt/vol) acrylamide gel (*Figure 4—figure supplement 3*). The protein sample was concentrated to 10 mg/ml and exchanged to a buffer containing 10 mM HEPES, pH 7.4, 50 mM KCl, 10 mM BME for storage at −80°C.

### Crystallization, data collection, and structure determination
Peptide synthesis of WT and mutant $PLN_{8-22}$ was performed by Genscript Biotech Corporation. The purities of the peptides were >98% as assessed by analytical high-performance liquid chromatography and their molecular masses were verified by ESI-MS. The $PKAc:AMP-PNP:PLN_{8-22}:Mg^{2+}$ complex was formed by combining a 1:10:10:10 molar ratio mixture of PKAc (6.5 mg/ml), AMP-PNP, $PLN_{8-22}$, and $MgCl_2$ in 10 mM HEPES (pH 7.4), 150 mM KCl, and 10 mM BME at room temperature for 5 min.

Initial crystallization screening was performed by the sitting-drop vapor-diffusion method using commercial crystal sparse matrix screen kits from Hampton Research and Molecular Dimensions. The crystal setting was carried out in 96-well format using a 1:1 ratio with an automated liquid handling robotic system (Gryphon, Art Robbins). After obtaining the initial hits, optimization of crystallization conditions was carried out using hanging-drop vapor diffusion in a 24-well format. The best crystallization condition for the complex with WT PLN contains 0.1 M BIS-TRIS, pH 6.5, and 25% [wt/vol] PEG 3350; the best condition for the complex with R9C PLN contains 0.1 M HEPES, pH 7.5, 0.2 M $MgCl_2$, and 25% PEG 3350; the best condition for the complex with A11E contains 0.1 M HEPES, pH 7.5, 0.2 M NaCl, and 25% PEG 3350. Crystals were mounted in Cryo-loops (Hampton Research) and

flash-cooled in liquid nitrogen with a reservoir solution containing 25% glycerol as cryoprotectant. Diffraction data were collected on BL17U1 at Shanghai Synchrotron Radiation Facility (SSRF) to resolutions of 2.4 Å ($PLN_{WT}$), 3.2 Å ($PLN_{R9C}$), and 2.8 Å ($PLN_{A11E}$), respectively. The dataset was indexed, integrated, and scaled using the HKL3000 suite (*Minor et al., 2006*). Molecular replacement was performed using the crystal structure of PKAc complexed with a 20-amino acid substrate analog inhibitor as a search model (PDB ID 2CPK) by PHENIX (*Adams et al., 2010*). After running Phaser-MR, we replaced the model sequence with the object sequences. The structure was further manually built into the modified experimental electron density using Coot (*Emsley and Cowtan, 2004*) and refined in PHENIX[57] in iterative cycles. The data collection and final refinement statistics are shown in *Table 1*. All structure figures were generated using UCSF Chimera (*Pettersen et al., 2004*).

## Determination of the oligomeric states of PKAc–PLN complexes

The oligomeric states of PKAc–PLN complexes were determined by gel-filtration chromatography. 0.1 mM PKAc protein was preincubated with 5 mM WT or mutant PLN peptides for 1 hr at 4°C before loaded on a Superdex 200 16/600 gel-filtration column (GE Healthcare) in buffer A. The column was calibrated using the gel filtration calibration kit (Sigma-Aldrich). Blue dextran ($M_R$ = 2000 kDa) was used to determine $V_0$. Thyroglobulin ($M_R$ = 669 kDa), apoferritin ($M_R$ = 443 kDa), β-amylase ($M_R$ = 200 kDa), alcohol dehydrogenase ($M_R$ = 150 kDa), albumin ($M_R$ = 66 kDa), and carbonic anhydrase ($M_R$ = 29 kDa) were used as protein standards. The predicted molecular weights of PKAc–PLN complexes were predicted using the plotted standard curve. Although the peptide concentrations in the samples were >sixfold higher than the $K_D$ value, which would probably make all peptides bound to PKA during the loading step, it is still possible that the peptide dissociates from PKA during the size exclusion run.

## Fluorescence-based thermal shift assays

The protein melting curves were measured using a fluorescence-based thermal shift assay (*Nettleship, 2008*). The Sypro orange dye (2×), PKAc (0.2 mg/ml), AMP-PNP (500 µM), and a PLN peptide variant (1 mM) were mixed in eight strip tubes (Axygen). The tubes were then transferred to a centrifuge and rotated to remove any bubbles and homogenize the system. The tubes were then placed into a Quant Studio 6 Flex real-time PCR machine (Life). The temperature was increased from 10 to 95°C with a ramping rate of 0.033°C/s. All measurements were performed in triplicate. The melting temperatures were obtained by taking the midpoint of each transition.

## ADP-Glo kinase assay

The kinase activity of PKAc was measured using the ADP-Glo kinase kit (V9101; Promega) according to the manufacturer's instructions. Phosphorylation of PLN peptides were performed at 30°C for 30 min in 50 µl kinase buffer (10 mM HEPES, pH 7.4, 150 mM KCl, 20 mM $MgCl_2$, 2 mM DTT) supplemented with 200 µM ATP, 10 nM PKAc, and 90 µM peptide substrates. 25 µl samples were removed and terminated by adding 25 µl ADP-Glo reagent followed by incubation at room temperature for 40 min. Kinase detection reagent was prepared by combining kinase detection buffer with kinase detection substrate based on the manufacturer's instructions. 50 µl kinase detection reagent was added and incubated at room temperature for 40 min to convert ADP to ATP. The luminescence signal was read by a Tecan Infinite M200 Pro plate reader. All measurements were performed in triplicate.

## SPR analysis

SPR experiments were carried out to characterize the interaction between PKAc and substrate peptides using a Biacore T200 instrument (GE Healthcare). PKAc was immobilized via standard *N*-hydroxysuccinimide (NHS)/1-ethyl-3-(3-dimethylaminopropyl) carbodiimide hydrochloride (EDC) amine coupling on a CM5 (carboxyl methyl dextran) sensor chip (GE Healthcare). Before covalent immobilization of PKAc, the sensor surface was activated by a mixed solution of 0.4 M EDC and 0.1 M NHS (1:1) for 7 min at a flow rate of 10 µl/min. The purified PKAc protein was diluted to 35 µg/ml in 200 µl of immobilization buffer (10 mM sodium acetate, pH 5.5) and immobilized on the sensor chip to a level of 7000 response units (RU). Interactions between PKAc and substrate peptides were monitored by injecting various concentrations of peptides (twofold serial dilutions starting from 1 or 2 mM) in the running buffer containing 10 mM HEPES, pH 7.4, 150 mM KCl, 20 mM $MgCl_2$, 1 mM AMP-PNP, and 0.005% (vol/vol) Surfactant P20 at a flow rate of 30 µl/min for 120 s. Dissociation was performed

by running the buffer without peptides at the rate of 30 μl/min for 120 s. The RU was obtained by subtracting a control for unspecific binding (the signal from a blank flow cell without PKAc subunit).

## NMR

The PLN peptides were dissolved in 10% or 100% $D_2O$. ROESY and TOCSY spectra were recorded at 298 K using an 850 MHz Bruker Avance NMR spectrometer equipped with a 5 mm cryogenic probe. NMR spectra were processed using NMRPipe (*Delaglio et al., 1995*) and analyzed using NMRFAM-Sparky (*Lee et al., 2015*). Distance constraints obtained from the assigned NOEs were divided into three classes based on the intensities of NOE crosspeaks: (1) strong: 1.8 Å $< d <$ 2.8 Å; (2) medium: 1.8 Å $< d <$ 3.4 Å; and (3) weak: 1.8 Å $< d <$ 5.5 Å. The solution structure was calculated with cyana 2.1 (*Güntert et al., 1997*). Twenty conformers from a total of 100 calculated ensembles with the lowest energy were selected for analysis.

## Acknowledgements

We thank J Xu from the Instrument Analytical Center of the School of Pharmaceutical Science and Technology at Tianjin University for assisting in using the in-house X-ray diffraction machine, J Shen at the Tianjin Institute of Industrial Biotechnology Chinese Academy of Sciences for assisting SPR analysis, and the staff at the beamline BL17U1 at Shanghai Synchrotron Radiation Facility. Funding: Funding for this research was provided by the National Natural Science Foundation of China (32022073 and 31972287, to Z.Y.), the Natural Science Foundation of Tianjin (19JCYBJC24500, to Z.Y.), CIHR (PJT-159601, to F.V.P.), and fellowships from the CIHR and Michael Smith Foundation for Health Research (to O.H.G.).

## Additional information

### Funding

| Funder | Grant reference number | Author |
|---|---|---|
| National Natural Science Foundation of China | 32022073 | Zhiguang Yuchi |
| National Natural Science Foundation of China | 31972287 | Zhiguang Yuchi |
| Natural Science Foundation of Tianjin City | 19JCYBJC24500 | Zhiguang Yuchi |
| Canadian Institutes of Health Research | PJT-159601 | Filip Van Petegem |
| Canadian Institutes of Health Research | Fellowship | Omid Haji-Ghassemi |
| Michael Smith Foundation for Health Research | Fellowship | Omid Haji-Ghassemi |

The funders had no role in study design, data collection, and interpretation, or the decision to submit the work for publication.

### Author contributions

Juan Qin, Conceptualization, Data curation, Formal analysis, Investigation, Visualization, Writing – original draft; Jingfeng Zhang, Lianyun Lin, Zhi Lin, Investigation; Omid Haji-Ghassemi, Filip Van Petegem, Investigation, Writing – review and editing; Kenneth J Woycechowsky, Data curation, Writing – review and editing; Yan Zhang, Investigation, Supervision, Writing – review and editing; Zhiguang Yuchi, Conceptualization, Formal analysis, Funding acquisition, Investigation, Resources, Supervision, Writing – original draft, Writing – review and editing

### Author ORCIDs
Juan Qin [iD] http://orcid.org/0000-0002-6762-3988
Zhiguang Yuchi [iD] http://orcid.org/0000-0003-2595-9106

Decision letter and Author response
Decision letter https://doi.org/10.7554/eLife.75346.sa1
Author response https://doi.org/10.7554/eLife.75346.sa2

## Additional files

### Supplementary files
• Transparent reporting form

### Data availability
Diffraction data have been deposited in PDB under the accession code: PKAc-WT PLN (PDB 7E0Z); PKAc-PLN R9C (PDB 7E11); PKAc-PLN A11E (PDB 7E12).

The following datasets were generated:

| Author(s) | Year | Dataset title | Dataset URL | Database and Identifier |
|---|---|---|---|---|
| Qin J, Yuchi Z | 2022 | Crystal structure of PKAc-PLN complex | https://www.rcsb.org/structure/7E0Z | RCSB Protein Data Bank, 7E0Z |
| Qin J, Yuchi Z | 2022 | Crystal structure of PKAc-PLN R9C complex | https://www.rcsb.org/structure/7E11 | RCSB Protein Data Bank, 7E11 |
| Qin J, Yuchi Z | 2022 | Crystal structure of PKAc-A11E complex | https://www.rcsb.org/structure/7E12 | RCSB Protein Data Bank, 7E12 |

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
