## [Editor Report]

As part of the "flight or fight" response, protein kinase A phosphorylates phospholamban (PLN), thereby relieving a tonic inhibition of the endo/sarco-plasmic reticulum calcium pump, which results in an increased force of cardiac contraction. This paper describes results on the association between protein kinase A and peptides corresponding to wild-type PLN (residues 8-22) and peptides bearing mutations (R9C and A11E) which in the context of full-length PLN (52 residues) result in dilated cardiomyopathy. The work will be of interest to investigators of mechanisms of substrate recruitment by protein kinases, and particularly to those who are trying to understand the mechanisms of familial dilated cardiomyopathy.

---

## [Decision Letter]

**Decision letter after peer review:**

Thank you for submitting your article "Structures of PKA-phospholamban complexes reveal a mechanism of familial dilated cardiomyopathy" for consideration by *eLife*. Your article has been reviewed by 3 peer reviewers, and the evaluation has been overseen by a Reviewing Editor and Richard Aldrich as the Senior Editor. The following individual involved in review of your submission has agreed to reveal their identity: Qing Yang (Reviewer #3).

Essential revisions:

Your manuscript has now been read by three reviewers whose comments are attached. They are in agreement that the work is well done and describes interesting new results on the association between PKA and peptides corresponding to wild-type PLN (residues 8-22) and peptides bearing mutations (R9C and A11E) which in the context of full-length PLN result in dilated cardiomyopathy. Although the referees do not believe that additional experiments are required, the consensus is that the manuscript needs to be re-written in such a way as to take into account the limitations of peptide studies for understanding the interaction between PKA and PLN in vivo, and for understanding the mechanism by which mutations result in DCM. I believe this is a valid concern and one that could in future studies be examined experimentally by, for example, testing how PLN phosphorylation is affected in hearts of mice with knock-in of DCM-causing mutations and determining how/if this relates to development of DCM.

*Reviewer #1 (Recommendations for the authors):*

Overall, this a well-executed study of significant import. Specific comments and suggestions for improvement of certain parts are itemized below.

The authors reference (39) – Ceholski et al. 2012 – as proposing a model in which PLN sequesters PKA. Reading this paper this is a little puzzling, as the authors of that paper note that pre-incubation with the mutant peptides (including R9C) did not inhibit phosphorylation with PKA and that these mutant peptides do not appear to directly sequester PKA in their assays (or maybe I misunderstood?). I think maybe the authors intended to reference Schmitt et al. 2003 (PMID 12610310) but I am not 100% sure.

I am a little unsure about how to interpret the thermal shift assay. Given that the peptide is present well in excess (1mM) of the Kd for all cases, I'm not sure how much small (~1-1.5degC) changes in Tm of the saturated complex tell us (as I presume the melting transition is dominated by unfolding of PKA, and it is not clear to me what the quantitative nature of the relationship between Tm and affinity might be in this case). It might be more informative to instead measure the melting curve for each complex at varying peptide concentration, to get a more quantitative estimate of affinity independent of tethering to the SPR chip (as I think this is the goal of this experiment).

Using the ADP-Glo assay, the authors show relatively mild effects of most of the mutants on PKA phosphorylation, in the context of isolated, monomeric peptides corresponding to the cytoplasmic tail of PLN. I think it might be worth briefly mentioning that in vivo, these trends may be significantly affected by the oligomerization state of PLN. For example, Ceholski et al. 2012 also measure phosphorylation of the R9C peptide and show that this mutation severely inhibits PKA phosphorylation in the context of full-length pentameric PLN, but has a much milder effect when monomeric, bound to SERCA, or as an isolated tail peptide.

Figure 4D is labeled as "relative activities of PKAc with different peptide substrates", but the y-axis is only labeled as "Luminescence", with no units. Perhaps it might be better to show as percentage of wild-type activity? Or otherwise some explanation of the units here would help readers understand what is going on.

The comparison with 3O7L (Figure 2 and associated discussion) I think could use a bit of refinement. I had a look at this structure and it is pretty clear that the B-factor has been incorrectly assigned (it is 2.0 (!), as opposed to the 45-50 values in the surrounding region…), and this is the likely cause of the negative density in the Fo-Fc map highlighted in Figure 2A. If one looks at the re-refined coordinates and map in the PDB-REDO database, it is pretty clear that the nucleotide and first two phosphates at least are present (I think the third is likely hydrolyzed, but it's not totally clear). Contouring the 2Fo-Fc omit maps at 3 σ for the nucleotide, and 1.5 σ for the gly-rich loop, is a little misleading in this regard – I calculated a simulated-annealing omit map myself for 3O7L and it is evident that there is a nucleotide there, even if the original B-factor assignment was incorrect. I would moderate discussion/display of this accordingly.

Please label the elution volumes on Figure S2 – it is a little difficult to interpret otherwise, as the expected change in elution volume for a dimer-monomer transition in this size range is pretty small on an S200 column (an S75 might be a better choice if available). There should also be a section in the methods outlining what calibration standard were used and the nature of the samples applied to the column. How do we know the peptide is co-eluting with PKA on gel filtration, given the low affinity of the complex?

*Reviewer #2 (Recommendations for the authors):*

None.

*Reviewer #3 (Recommendations for the authors):*

I enjoy reading this well-written manuscript and appreciate the great contribution by the authors to the pathological mechanism of PLN-related DCM. I suggest accepting the manuscript after a revision of following problems.

1. The authors could consider presenting figure 2 as supplementary data since it mainly provides evidence of the discrepancy between the published structure of PKA-PLN complex and their newly determined high-resolution crystal structure. It is already clear that the original structure was not properly determined. This has been clearly illustrated by figure 5 and stated by the text.

2. Line 277: "Nontheless" should be "Nonetheless".

3. Line 25: The origin of the protein should be mentioned.

4. Line 58-59, 63-64: references should be included.

5. Line 293-295: "In all other available complex structures of PKAc", references should be added.

6. Line 335: "(figure 3E, 4)" should be modified to "(figure 3E, 4C, 4D)" to be more accurate.

7. Line 359: there is no pSer16 presented in Figure 4C. "4C" may be corrected as "4B".

8. Line 511-512: "These errors in the previous model would certainly impair our understanding of the mechanism by which PKA regulates PLN." This description seems too strict.

9. Line 543: gene accession numbers should be present. If genes were not synthesized by corporation, a complete list of all primers used in this study should be supplied in Supplementary materials.

10. The Ramachandran outliers should be less than 0.2 %.

---

## [Author Response]

Reviewer #1 (Recommendations for the authors):Overall, this a well-executed study of significant import. Specific comments and suggestions for improvement of certain parts are itemized below.The authors reference (39) – Ceholski et al. 2012 – as proposing a model in which PLN sequesters PKA. Reading this paper this is a little puzzling, as the authors of that paper note that pre-incubation with the mutant peptides (including R9C) did not inhibit phosphorylation with PKA and that these mutant peptides do not appear to directly sequester PKA in their assays (or maybe I misunderstood?). I think maybe the authors intended to reference Schmitt et al. 2003 (PMID 12610310) but I am not 100% sure.

We thank the reviewer for identifying the mistake. We have corrected the citation by removing the reference of Ceholski et al. 2012 and adding the new reference of Schmitt et al. 2003.

I am a little unsure about how to interpret the thermal shift assay. Given that the peptide is present well in excess (1mM) of the Kd for all cases, I'm not sure how much small (~1-1.5degC) changes in Tm of the saturated complex tell us (as I presume the melting transition is dominated by unfolding of PKA, and it is not clear to me what the quantitative nature of the relationship between Tm and affinity might be in this case). It might be more informative to instead measure the melting curve for each complex at varying peptide concentration, to get a more quantitative estimate of affinity independent of tethering to the SPR chip (as I think this is the goal of this experiment).

We agree with the reviewer that the current results of the thermal shift assay does not reflect the quantitative percentage binding considering the peptides were given in excess. Instead, the observed difference in Tm is likely due to the difference in the interaction mode between the peptides and PKA. The basis of this experiment is that ligand binding stabilizes the tertiary structure of the protein, and the tighter the binding (higher affinity, lower Kd), the greater the stabilization of the protein’s tertiary structure. By measuring the difference in Tm of PKA between unbound and ligand-saturated conditions, we can compare the extent to which the tertiary structure of PKA has been stabilized between the ligand variants, which should reflect their relative affinities. Mutant PLN has less interactions with PKA compared to WT PLN so that the stabilizing effect should be also less. The results of the thermal shift assay gave an indirect evidence the mutation reduces the interactions between PLN and PKA, which in turn destabilizes the complex slightly. Due to the poor resolution of the assay, it is probably difficult to observe further subtle difference if we redo the assay using varying peptide concentrations. After all, thermal shift is more a qualitative than a quantitative assay. It was used only to support the main conclusion from the more quantitative SPR results. We have added a statement in the result section “Considering the peptides were given in excess, the results of the thermal shift assay does not reflect the quantitative percentage binding but rather the difference in the interaction mode between the peptides and PKA.”

Using the ADP-Glo assay, the authors show relatively mild effects of most of the mutants on PKA phosphorylation, in the context of isolated, monomeric peptides corresponding to the cytoplasmic tail of PLN. I think it might be worth briefly mentioning that in vivo, these trends may be significantly affected by the oligomerization state of PLN. For example, Ceholski et al. 2012 also measure phosphorylation of the R9C peptide and show that this mutation severely inhibits PKA phosphorylation in the context of full-length pentameric PLN, but has a much milder effect when monomeric, bound to SERCA, or as an isolated tail peptide.

We thank the reviewer for the great suggestion. We have added this point to the Discussion section “Additionally, the trend of the reduced phosphorylation by DCM mutations can be significantly affected by the oligomerization state of PLN. Ceholski et al. showed that R9C severely inhibits PKA phosphorylation in the context of full-length pentameric PLN, but has a much milder effect in the context of full-length monomeric PLN or an isolated tail peptide [41].”

Figure 4D is labeled as "relative activities of PKAc with different peptide substrates", but the y-axis is only labeled as "Luminescence", with no units. Perhaps it might be better to show as percentage of wild-type activity? Or otherwise some explanation of the units here would help readers understand what is going on.

Thanks for the suggestion. We have changed the Y-axis of the panel D to the normalized luminescence values using WT PLN as control, which could better reflect the relative activities of different substrates.

The comparison with 3O7L (Figure 2 and associated discussion) I think could use a bit of refinement. I had a look at this structure and it is pretty clear that the B-factor has been incorrectly assigned (it is 2.0 (!), as opposed to the 45-50 values in the surrounding region…), and this is the likely cause of the negative density in the Fo-Fc map highlighted in Figure 2A. If one looks at the re-refined coordinates and map in the PDB-REDO database, it is pretty clear that the nucleotide and first two phosphates at least are present (I think the third is likely hydrolyzed, but it's not totally clear). Contouring the 2Fo-Fc omit maps at 3 σ for the nucleotide, and 1.5 σ for the gly-rich loop, is a little misleading in this regard – I calculated a simulated-annealing omit map myself for 3O7L and it is evident that there is a nucleotide there, even if the original B-factor assignment was incorrect. I would moderate discussion/display of this accordingly.

Thanks for the careful inspection and reprocessing of the deposited structural data. Indeed, the results of PDB-REDO support the presence of the nucleotide in the original structure but the PEG molecule was probably still misplaced in the location of the gly-rich loop. We have added new panels in the Figure 2 to show the results from PDB-REDO, changed the contour levels of both 2Fo-Fc omit maps to 2, removed the panel B showing the binding site colored by B-factors, and moved this figure to the supplementary material (new Figure 4—figure supplement 1). We have also removed all the content about B-factor from the paper, and revised the result section to “A further difference lies in the active site. Our model shows clear electron density for AMP-PNP and the nearby glycine-rich loop (gly-loop) (Figure 4—figure supplement 1, Figure 1—figure supplement 1). In 3O7L, the γ-phosphate of AMP-PNP was not modeled, and there is a clear negative difference density for the rest of molecule according to the map generated using the previously deposited structure factor data, reflecting a low occupancy of the nucleotide (Figure 4D, Figure 4—figure supplement 1). The re-refined coordinates and map in the PDB-REDO database are improved compared to the original 3O7L, showing that the nucleotide and first two phosphates are present while the γ-phosphate is likely hydrolyzed (Figure 4—figure supplement 1). Further, the 3O7L structural model contains a PEG molecule that is located in a patch of negative difference density, which raises the question of whether or not it is actually present. Neighboring positive difference density is more likely to correspond to the gly-loop according to the comparison of the two structures (Figure 4C,D, Figure 4—figure supplement 1). The gly-loop is right next to the nucleotide and is crucial for nucleotide coordination. Thus, it would be very unusual for a PEG molecule to occupy this loop position, especially when the nearby catalytic loop residues Asp166 and Lys168 are in similar positions in both structures.” We also revised the Discussion section to “Second, the electron density map of 3O7L has poor quality in the regions of AMP-PNP and PEG.”

Please label the elution volumes on Figure S2 – it is a little difficult to interpret otherwise, as the expected change in elution volume for a dimer-monomer transition in this size range is pretty small on an S200 column (an S75 might be a better choice if available). There should also be a section in the methods outlining what calibration standard were used and the nature of the samples applied to the column. How do we know the peptide is co-eluting with PKA on gel filtration, given the low affinity of the complex?

We have added the elution volumes on new figure, Figure 4—figure supplement 3. The expected change in elution volume for a dimer-monomer transition is ~10 mL on our Superdex200. It is small but enough to distinguish dimer from monomer. It was not ideal to use Superdex200 but that is what we have got in the laboratory. We have added a new section in method to summarize the detailed experimental procedure.

“Determination of the oligomeric states of PKAc-PLN complexes

The oligomeric states of PKAc-PLN complexes were determined by gel-filtration chromatography. 0.1 mM PKAc protein was pre-incubated with 5 mM WT or mutant PLN peptides for 1 hour at 4 ℃ before loaded on a Superdex 200 16/600 gel-filtration column (GE Healthcare) in buffer A. The column was calibrated using the gel filtration calibration kit (Σ Aldrich). Blue dextran (M_R_ = 2,000 kDa) was used to determine V_0_. Thyroglobulin (M_R_ = 669 kDa), apoferritin (M_R_=443 kDa), β-Amylase (M_R_=200 kDa), alcohol dehydrogenase (M_R_=150 kDa), albumin (M_R_=66 kDa), and carbonic anhydrase (M_R_=29 kDa) were used as protein standards. The predicted molecular weights (MWs) of PKAc-PLN complexes were predicted using the plotted standard curve. Although we used concentrations 10x higher than the Kd, which would probably make all peptides bound to PKA during the loading step, it is still possible that the peptide dissociates from PKA during the size exclusion run.”

The concentrations of peptides we used for SEC were over 6-fold higher than the K_D_ value. Thus, we suppose that the peptides should bind with PKA, at least at the loading step, although technically the complexes could still fall apart during the run. We have added a statement to the method section

“Although the peptide concentrations in the samples were >6-fold higher than the K_D_ value, which would probably make all peptides bound to PKA during the loading step, it is still possible that the peptide dissociates from PKA during the size exclusion run.”

Reviewer #3 (Recommendations for the authors):I enjoy reading this well-written manuscript and appreciate the great contribution by the authors to the pathological mechanism of PLN-related DCM. I suggest accepting the manuscript after a revision of following problems.1. The authors could consider presenting figure 2 as supplementary data since it mainly provides evidence of the discrepancy between the published structure of PKA-PLN complex and their newly determined high-resolution crystal structure. It is already clear that the original structure was not properly determined. This has been clearly illustrated by figure 5 and stated by the text.

We have moved Figure 2 to the supplementary data as the reviewer suggested.

2. Line 277: "Nontheless" should be "Nonetheless".

We have corrected it.

3. Line 25: The origin of the protein should be mentioned.

We have added the origin “mouse” to the abstract and the method.

4. Line 58-59, 63-64: references should be included.

We have added the relevant references at both locations.

5. Line 293-295: "In all other available complex structures of PKAc", references should be added.

We have added the relevant references.

6. Line 335: "(figure 3E, 4)" should be modified to "(figure 3E, 4C, 4D)" to be more accurate.

We have corrected it accordingly.

7. Line 359: there is no pSer16 presented in Figure 4C. "4C" may be corrected as "4B".

Thanks for identifying the error. The citation should be “4D” (new Figure 3D). We have corrected it.

8. Line 511-512: "These errors in the previous model would certainly impair our understanding of the mechanism by which PKA regulates PLN." This description seems too strict.

We have changed it to “These errors in the previous model would certainly compromise our understanding of the mechanism by which PKA regulates PLN."

9. Line 543: gene accession numbers should be present. If genes were not synthesized by corporation, a complete list of all primers used in this study should be supplied in Supplementary materials.

We have added the gene accession number to the method section.

10. The Ramachandran outliers should be less than 0.2 %.

The sums of the favored and the allowed for WT and A11E are both 100%. The one for R9C is over 99.7%, a little short of 99.8% due to the low resolution of the dataset.